# Bridging Horizons: Exploring STEM Students' Perspectives on Service-Learning and Storytelling Activities for Community Engagement and Gender Equality

Cristina Tripon 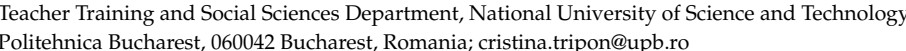

Teacher Training and Social Sciences Department, National University of Science and Technology Politehnica Bucharest, 060042 Bucharest, Romania; cristina.tripon@upb.ro

**Abstract:** This study explores STEM students' perspectives on service-learning and story-telling activities to enhance community engagement and advance gender equality, investigating their impact on students' perceptions, experiences, and understanding of gender dynamics within rural communities. Through qualitative analysis of interviews, reflective journals, and participatory videos, this study explores the transformative potential of service-learning and storytelling initiatives in empowering rural women, challenging traditional societal roles, and advocating for equal opportunities, particularly in STEM disciplines. Findings reveal the multifaceted benefits of these activities, including the development of empathy, cultural awareness, leadership skills, and a commitment to social justice among participating students. This study highlights the importance of integrating service-learning and storytelling into STEM education to cultivate inclusive practices, promote community development, and advance gender equality in rural settings.

**Keywords:** gender equality; STEM education; service-learning; storytelling

## 1. Introduction

In recent years, there has been growing recognition of the importance of fostering STEM education in rural communities to bridge the educational divide and empower future generations. This research seeks to explore how STEM students, as future educators, perceive and engage with service-learning experiences in rural contexts, and how these experiences shape their understanding of inclusive teaching practices.

In the realm of education, fostering inclusive teaching practices is paramount for creating learning environments where all students thrive [1,2]. Rural communities, often facing unique socio-economic and cultural challenges, require educators equipped with a nuanced understanding of inclusivity. Service-learning is a transformative pedagogy that engages students, community members, and instructors in co-creating partnerships to address community challenges. Service-learning emerges as a promising avenue for STEM students to engage with rural communities, confront real-world educational issues, and cultivate inclusive teaching strategies [3].

## 2. Gender Equality, Service-Learning, STEM Approaches, and Storytelling

Achieving gender equality in rural communities is not just a matter of social justice; it is a fundamental necessity for sustainable development, economic growth, and the realization of human rights.

Gender equality is a fundamental human right enshrined in international agreements such as the Universal Declaration of Human Rights [4]. All individuals, regardless of gender, have the right to live free from discrimination, violence, and oppression. Achieving gender equality ensures that women and girls in rural communities have equal access to opportunities, resources, and decision-making processes, enabling them to lead fulfilling lives with dignity and respect [5–7].

Gender inequality exacerbates poverty and hinders economic development in rural areas. Women and girls often face limited access to education, healthcare, land ownership, and financial resources, which restricts their ability to participate in the economy and improve their living standards [8]. Closing the gender gap in rural communities by providing equal opportunities for education, training, and economic empowerment not only benefits women and girls but also contributes to poverty reduction, economic growth, and sustainable development.

Women play a crucial role in agricultural production and food security in rural communities, yet they often lack access to land, credit, technology, and extension services [9]. Achieving gender equality in agriculture and rural development is essential for increasing productivity, improving livelihoods, and ensuring food security for households and communities. Empowering women farmers through training, access to resources, and participation in decision-making processes can enhance agricultural productivity, promote sustainable land management practices, and strengthen rural economies [10].

Gender inequality undermines the health and well-being of women and girls in rural communities. Limited access to healthcare services, information, and reproductive rights often results in higher maternal mortality rates, lower life expectancy, and greater vulnerability to diseases such as HIV/AIDS [11]. Promoting gender equality in healthcare delivery, education, and decision-making can improve health outcomes, reduce disparities, and enhance the overall well-being of rural populations.

Education is a powerful tool for empowering women and girls, challenging gender stereotypes, and promoting social change [12]. Yet, in many rural areas, girls face barriers to accessing quality education, including poverty, cultural norms, child marriage, and lack of infrastructure. Achieving gender equality in education requires addressing these barriers and ensuring equal opportunities for girls to learn, thrive, and fulfill their potential [13,14]. Education empowers women and girls to make informed choices, participate in decision-making processes, and contribute to the social, economic, and political development of their communities It requires concerted efforts and collective action from governments, civil society, the private sector, and communities to address the root causes of gender inequality, dismantle discriminatory norms and practices, and create an enabling environment where all individuals can realize their rights and aspirations, irrespective of their gender or background [15–17].

The STEM approach, which stands for Science, Technology, Engineering, and Mathematics, is an interdisciplinary method of education that integrates these four key disciplines into a cohesive learning paradigm.

At its core, the STEM approach fosters an inquiry-based learning environment where students engage in hands-on activities, experiments, and projects that mirror real-world scenarios. Rather than learning subjects in isolation, students are encouraged to apply concepts from science, technology, engineering, and mathematics to solve complex problems and explore the interconnectedness of these disciplines.

This approach (Figure 1) not only promotes a deeper understanding of core concepts but also cultivates essential skills such as communication, teamwork, and adaptability. STEM education breaks down the traditional silos between subjects, allowing students to see the connections between science, technology, engineering, and mathematics. Students engage in hands-on activities, experiments, and projects that encourage active learning and the application of theoretical concepts. STEM education emphasizes problem-solving skills by presenting students with real-world challenges that require critical thinking and creativity to solve [18–20]. Technology is integrated seamlessly into the STEM curriculum, enabling students to leverage tools and resources to collect data, analyze information, and design solutions. STEM encourages collaboration and teamwork, reflecting the interdisciplinary nature of modern workplaces where individuals with diverse skill sets must work together to achieve common goals. STEM education nurtures creativity and innovation by empowering students to explore multiple solutions to problems and think outside the box [21–23]. STEM learning experiences are designed to be relevant and applicable to

real-world contexts, preparing students for future careers in STEM fields and equipping them with the skills needed to address global challenges.

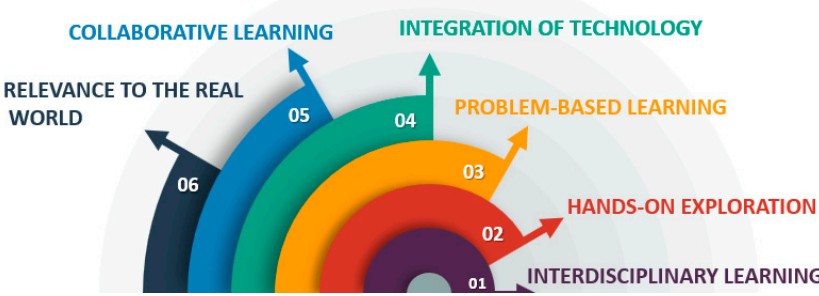

**Figure 1.** STEM approach.

The STEM approach represents a paradigm shift in education, moving away from rote memorization and passive learning towards a dynamic and engaging model that empowers students to become lifelong learners and active contributors to society [24].

The connection between the storytelling approach and the STEM (Science, Technology, Engineering, and Mathematics) approach lies in how storytelling can enhance engagement, understanding, and retention of STEM concepts among students. Storytelling in STEM education serves as a powerful pedagogical tool that makes complex scientific information more approachable, relatable, and meaningful to learners, from early ages [25,26]. By incorporating narratives, anecdotes, and real-world examples into STEM lessons, educators can create educational environments where students develop emotional connections to the subject matter, fostering deeper engagement and interest in STEM disciplines.

Research and articles [24–26] highlight the benefits of using storytelling in STEM education to communicate science effectively, engage students in problem-solving activities, and enhance their critical thinking skills. Storytelling not only makes STEM content more accessible but also helps students connect theoretical knowledge to practical applications, promoting a deeper understanding of scientific concepts [25,26].

The integration of storytelling in STEM education aligns with the goal of making STEM learning more engaging, relevant, and impactful for students. By weaving narratives into STEM instruction [27], educators can create a dynamic learning environment that fosters creativity, critical thinking, and a deeper appreciation for the interconnectedness of science, technology, engineering, and mathematics in real-world contexts.

Storytelling is a timeless and captivating approach to conveying ideas, emotions, and experiences through the art of narrative. It transcends cultures, languages, and generations, connecting individuals through the power of shared imagination and empathy. At its core, storytelling involves crafting a compelling narrative that engages, entertains, and resonates with its audience. The storytelling approach encompasses several key elements [28]:

✓ Narrative Structure: A well-structured story typically follows a beginning, middle, and end format, often referred to as the three-act structure. This framework allows for the introduction of characters, setting the stage for conflict or tension, development of plot and characters, and ultimately, resolution or climax.

✓ Characters: Memorable characters are central to effective storytelling. They serve as the conduits through which audiences engage with the narrative, empathize with struggles, and celebrate triumphs. Well-developed characters possess depth, complexity, and relatability, making them compelling and memorable.

✓ Emotional Resonance: Storytelling evokes emotions ranging from joy and laughter to sorrow and fear. By tapping into universal themes and experiences, storytellers can create emotional resonance that fosters empathy and connection with their audience.

✓ Imagery and Descriptive Language: Vivid imagery and descriptive language bring stories to life, transporting audiences to different times, places, and worlds. By ap-

pealing to the senses, storytellers immerse their audience in rich, sensory experiences that enhance engagement and immersion.

✓ Themes and Messages: Stories often convey deeper themes, moral lessons, or messages that resonate with audiences on a personal or universal level. Whether exploring themes of love, friendship, courage, or the human condition, storytelling provides a platform for reflection, introspection, and understanding.

✓ Engagement and Interactivity: In contemporary settings, storytelling can extend beyond traditional mediums such as books and oral storytelling to include interactive experiences, such as digital storytelling, immersive theater, and interactive media. These platforms encourage audience participation, collaboration, and exploration, blurring the lines between storyteller and audience.

The storytelling approach is a dynamic and versatile tool for communication, entertainment, and cultural expression. It celebrates the diversity of human experience, fosters empathy and understanding, and invites audiences on a journey of imagination and discovery [29,30]. Whether shared around a campfire, depicted on a stage, or experienced through digital media, storytelling continues to captivate hearts and minds, leaving an indelible mark on the fabric of human history and culture.

Service-learning is an educational approach that integrates meaningful community service with academic learning, aiming to foster civic responsibility and personal growth among participants. In this approach, students engage in service activities that address real community needs while simultaneously reflecting on their experiences to gain deeper insights and understanding [31–33].

At its core, service-learning goes beyond traditional classroom instruction by actively involving students in hands-on projects that have tangible benefits for both the community and the learners themselves. These projects can encompass a wide range of issues, including environmental conservation, social justice, healthcare access, education, poverty alleviation, and more. Service-learning approaches include [34–36]:

✓ Community Engagement: Service-learning emphasizes collaboration with community partners to identify needs, develop projects, and implement solutions. By working closely with local organizations, schools, nonprofits, and government agencies, students gain a deeper understanding of community dynamics and challenges.

✓ Academic Integration: Service-learning activities are intentionally linked to academic curriculum and learning objectives. Students apply classroom knowledge and skills to real-world situations, enhancing their understanding of course content while also developing practical skills such as critical thinking, problem-solving, communication, and teamwork.

✓ Reflection and Critical Thinking: Reflection is a central component of service-learning. Through guided reflection exercises, students examine their experiences, explore the social, cultural, and ethical dimensions of their service work, and critically analyze the impact of their actions on themselves and the community. Reflection helps students connect theory with practice, develop empathy and cultural competence, and identify areas for personal and academic growth.

✓ Civic Responsibility and Social Justice: Service-learning promotes values of civic engagement, social responsibility, and ethical leadership. By actively participating in community service, students develop a sense of agency and empowerment, become more aware of social issues and inequalities, and are inspired to take action to create positive change in their communities and beyond.

✓ Long-Term Impact and Sustainability: Service-learning aims to create lasting partnerships and sustainable solutions to community problems. By engaging in ongoing service projects and building relationships with community stakeholders, students contribute to meaningful change while also learning about the complexities of social issues and the importance of long-term commitment and collaboration.

Service-learning offers a transformative educational experience that empowers students to become active citizens, critical thinkers, and compassionate leaders who are committed to making a positive difference in the world.

## 3. Research Design

The Storytelling Club is a service-learning initiative aimed at challenging traditional societal roles and empowering women from rural communities to become active participants in shaping their futures beyond traditional roles such as housekeeping. The club brings together women of diverse ages (from 19–46 years) and backgrounds to share their stories, experiences, and aspirations through storytelling sessions. The primary objective of the club is to provide a space for rural women to amplify their voices, share their experiences, and explore alternative pathways beyond conventional gender roles. By fostering a supportive and inclusive environment, the club aims to empower women to pursue educational and career opportunities, engage in community development initiatives, and actively participate in their children's education. The students engaged in a variety of activities that integrated STEM principles and concepts within the context of service-learning as storytelling sessions and skill-building workshops. Regular storytelling sessions were held where women shared personal narratives, anecdotes, and life experiences. These sessions provided a space for self-expression, reflection, and mutual support among participants. The club organized workshops focused on developing practical skills such as public speaking, writing, and digital literacy. These workshops aimed to enhance participants' confidence and empower them to articulate their thoughts and ideas effectively.

The aim of this research was to examine the influence of service-learning experiences on STEM students' perspectives and readiness for future teaching positions in rural communities. Through qualitative analysis, this study delved into the reflections, insights, and transformative encounters of STEM students involved in service-learning projects designed to tackle educational obstacles in rural environments.

The recruitment process began with targeted outreach to university STEM programs known for their emphasis on teacher preparation and community engagement. These programs include undergraduate degrees in education, STEM disciplines, or interdisciplinary studies that integrate both fields.

Research questions:

1. What is the impact of service-learning experiences on STEM students' attitudes, beliefs, and preparedness for future teaching roles in rural communities?
2. How do service-learning initiatives in rural settings shape the perspectives on teaching and community engagement of STEM students, as reflected in their insights and reflections?

*Participants*

The recruitment of participants from university STEM programs with a focus on teacher preparation and community engagement in a research design offered a unique opportunity to investigate the intersection of academic learning, practical experience, and community impact.

Participation in the research study was voluntary, and all thirty-three participants (twenty-one students and twelve women) were provided with detailed information about the study objectives, procedures, risks, and benefits before consenting to participate. Informed consent forms ensured that all participants understood their rights, confidentiality measures, and the voluntary nature of their involvement in the research.

The women participants in the storytelling club belonged to a broad spectrum of ages, ranging from 19 to 46 years old. The twelve members of this category of participants contributed to a rich exchange of perspectives (diversity in age), with members representing different life stages, viewpoints, experiences, and the community diversity.

In the research study, twenty-one STEM student participants engaged in storytelling activities as part of the research methodology. These students were actively involved in various STEM disciplines, spanning science, technology, engineering, and mathematics.

The research was conducted in a rural area, Calarasi county, Curcani, Romania. Twelve women from this area and their children were the target group of the service-learning activities carried out by students from the Polytechnic University of Bucharest. The community of Curcani was selected to participate in the service-learning activity based on several factors, with the primary consideration being the expressed interest and advocacy from one of the STEM students who hailed from this community. This student, who was intimately familiar with the challenges and needs of Curcani, voiced a strong desire to contribute to the betterment of their hometown and to raise awareness of its necessities.

Moreover, Curcani presented characteristics that aligned with the objectives of the service-learning activity, such as being a rural community facing socio-economic challenges and gender disparities. Additionally, the presence of local stakeholders, such as educators, community leaders, and non-profit organizations, provided opportunities for meaningful collaboration and engagement in service-oriented projects.

*Research Method*

A qualitative approach, utilizing semi-structured interviews, participatory video, and reflective journals, offers a rich and nuanced method to capture the experiences, perceptions, and insights of STEM students participating in service-learning activities within rural communities.

Semi-structured interviews were used in this research and combined the flexibility of open-ended questions with the structure of a predetermined interview guide. This approach allowed us to explore specific topics while also allowing for in-depth exploration of participants' responses [37], detailed insights, capturing diverse perspectives, and generating nuanced understandings of complex educational phenomena. By engaging students in semi-structured interviews, we could gather rich qualitative data about the impact of service-learning on inclusive teaching practices, community engagement, and STEM education in underserved regions.

Semi-structured interviews provided a flexible yet focused framework for exploring participants' experiences and perspectives in depth. These interviews allowed participants to share their stories, insights, challenges, and transformative moments related to their service-learning experiences in rural communities. Also incorporated were reflective journals as a complementary method for capturing participants' ongoing thoughts, reflections, and experiences throughout their service-learning journey, and participants were encouraged to document their observations, insights, challenges, and growth experiences in written form, allowing for a deeper exploration of their internal processes and learning trajectories over time.

Reflective journaling promotes self-reflection and critical thinking among participants, encouraging them to examine their assumptions, biases, and learning edges [37,38]. By engaging in reflective practice, participants gain insights into their own beliefs, values, and teaching philosophies, as well as their evolving understanding of inclusive teaching practices and social justice issues within rural education (Table 1—Examples of interview items).

Participatory video was a research method used that involved collaborating with participants to create videos that captured their perspectives, experiences, and insights. This approach empowered participants to actively engage in the research process, share their stories, and contribute to knowledge creation. Videos conveyed emotions, contexts, and complexities that might have been challenging to capture through traditional research methods, enhancing the richness of data [39].

The combination of these research methods allowed for a holistic examination of the effectiveness of service-learning, providing diverse data sources and perspectives. By incorporating reflective practices and participatory approaches, the research process became inclusive, empowering participants to contribute to the narrative and share their experiences authentically. Through a triangulation of data from semi-structured interviews, reflective journals, and participatory videos, researchers could uncover nuanced insights that informed the development of inclusive teaching practices, community partnerships, and STEM education initiatives in underserved regions.

**Table 1.** Examples of interview items.

| | |
|---|---|
| Section 1: Reflection on Service-Learning Activities | ✓ Can you briefly describe the service-learning activities you were involved in?<br>✓ What motivated you to participate in these activities?<br>✓ How do you believe these activities align with the broader goal of emphasizing the role of women in society and promoting equal opportunities, especially in STEM? |
| Section 2: Learning and Skill Development | ✓ In what ways do you feel these service-learning activities contributed to your personal and academic growth?<br>✓ Were there any specific skills related to gender equality, teamwork, or community engagement that you believe you developed during the service-learning experience? |
| Section 3: Impact on Perceptions and Attitudes | ✓ Before participating in these activities, how did you perceive the role of women in STEM fields?<br>✓ Have your perceptions changed because of the service-learning experience?<br>✓ In your opinion, how can service-learning initiatives contribute to challenging stereotypes and fostering inclusivity in STEM disciplines? |
| Section 4: Community Interaction and Engagement | ✓ How did the service-learning activities facilitate interaction with the local community, especially in relation to gender equality?<br>✓ Did you observe any specific instances where the community benefited from the service-learning projects, particularly in terms of promoting equal opportunities for women? |
| Section 5: Recommendations and Future Involvement | ✓ Based on your experience, what recommendations would you provide for improving future service-learning initiatives focused on gender equality and equal opportunities in STEM?<br>✓ Do you see yourself continuing to be involved in similar community-oriented initiatives in the future? Why or why not? |
| Section 6: Personal Connection and Empathy | ✓ Can you share a specific experience or interaction during the service-learning activities that had a significant impact on your understanding of the challenges faced by women in STEM fields?<br>✓ In what ways has this experience influenced your personal commitment to promoting gender equality and equal opportunities in your academic and professional journey? |

## 4. Results

To assess the impact of service-learning experiences on STEM students' attitudes and preparedness for teaching roles in rural communities, a comprehensive analysis plan was devised. This plan included conducting semi-structured interviews with students to explore their reflections on service-learning activities, learning and skill development, the impact on perceptions and attitudes, community interaction and engagement, recommendations and future involvement, and personal connection and empathy. Reflective journaling was utilized to capture personal and professional growth insights. Additionally, participatory video creation was facilitated to showcase their experiences and perspectives.

Furthermore, the reflections and insights of STEM students engaged in service-learning initiatives in rural settings were investigated to provide valuable information on how these experiences shaped their views on teaching and community engagement.

For qualitative analysis, thematic analysis was used to identify key themes emerging from interviews, journal entries, and participatory videos. This involved looking for patterns in attitudes, beliefs, and preparedness for teaching roles, as well as comparing and contrasting reflections to understand the impact of service-learning on students' perspectives.

For cross-data validation, findings from interviews, journal entries, and videos were triangulated to validate themes and insights. Consistency and discrepancies across all data sources were examined to enhance the credibility of the analysis. One way to navigate subjectivity was by having members of the research team discuss the analysis and patterns to improve interpretation.

Triangulation was used as a methodological approach that involves the systematic integration of data from multiple sources to enhance the credibility and validity of qualitative findings within a specific educational context like service-learning. By incorporating diverse research methods (semi-structured interviews, reflective journals, participatory

video) and data collection techniques, triangulation served to provide a more comprehensive and reliable analysis of the subject under investigation. This enabled the researchers to triangulate different perspectives, data sources, and methodologies to strengthen the trustworthiness of their interpretations and deepen their understanding of complex educational phenomena.

In the research context, this study was conducted at the largest STEM university in the country, situated in the heart of a bustling cosmopolitan city. This university attracts students from a wide array of backgrounds and cultures. The selected students were well-accustomed to the fast-paced urban lifestyle prevalent in the city, characterized by towering skyscrapers, bustling nightlife, and a rich tapestry of languages and traditions. Most of these students had spent their formative years in urban environments, navigating the intricacies of city life without significant exposure to rural communities. Through service-learning experiences, students stepped out of the confines of the university campus and into rural educational settings. They encountered firsthand the challenges and opportunities present in rural schools, gaining practical insights into the realities of teaching in these communities.

In rural areas, students encountered a rich tapestry of cultures, socio-economic backgrounds, and linguistic diversity that differed from their urban upbringing. This exposure broadened their perspectives and fostered a deeper understanding of the unique needs and assets of communities.

Engaging in service-learning projects required collaboration and teamwork among STEM students, educators, and community members. This collaborative approach fostered a sense of camaraderie and collective responsibility towards addressing educational disparities in rural areas.

Service-learning prompted students to reflect critically on their own assumptions, biases, and preconceptions about rural education and community engagement. Through introspection and dialogue, students challenged their existing beliefs and cultivated a more nuanced understanding of inclusive teaching practices.

As students navigated the complexities of rural settings, they were compelled to adapt their pedagogical approaches to meet the diverse needs of their students. This involved implementing innovative teaching methods, leveraging technology, and incorporating culturally relevant content into their lesson plans.

Through service-learning experiences, students developed a heightened awareness of social injustices and inequities in education. They were motivated to advocate for systemic change and to leverage their STEM expertise to create more inclusive and equitable learning environments in rural communities.

Implementing activities of a service-learning and storytelling project in the life of a school community, and using community experiences, offered numerous benefits for STEM students.

- ○ Shaping STEM students' understanding of inclusive teaching practices in rural education
- ○ SL immerses STEM students in real-world educational contexts.
- ○ SL exposes STEM students to the cultural, socio-economic, and linguistic diversity inherent in rural communities.
- ○ SL encourages STEM students to collaborate.
- ○ SL encourages STEM students to engage in reflective practice and critical inquiry, examining their assumptions, biases, and learning edges.
- ○ SL prompts STEM students to develop adaptive pedagogical strategies.
- ○ SL instills in STEM students a sense of social justice and advocacy, inspiring them to become agents of change within their communities.

Preliminary findings indicated that service-learning experiences played a pivotal role in shaping STEM students' understanding of inclusive teaching practices in rural education. Participants expressed a heightened awareness of the socio-economic disparities and cultural nuances inherent in rural communities, underscoring the importance of con-

textually relevant pedagogy. Engaging directly with local stakeholders fostered empathy, cultural competency, and a deep appreciation for the assets and challenges present in rural educational settings, as students recorded in their reflective journals.

> *"I never fully appreciated the complexities of teaching in rural areas until I participated in a service-learning experience. Meeting with local educators, parents, and community leaders opened my eyes to the unique challenges they face. I realized that effective teaching in these settings requires more than just subject knowledge; it demands an understanding of the community's culture, socioeconomic dynamics, and resources. Through this experience, I've gained a deeper appreciation for the importance of contextually relevant pedagogy and the need for inclusive teaching practices"* (student M.).

> *"Before engaging in service-learning in rural education, I didn't fully grasp the extent of socio-economic disparities and cultural nuances present in these communities. Meeting with families, I realized the significance of tailoring teaching approaches to meet their needs and honor their backgrounds. It's not just about delivering content; it's about building relationships, understanding local contexts, and fostering a sense of belonging. Service-learning has helped me develop empathy, cultural competency, and a genuine appreciation for the assets and challenges present in rural educational settings"* (student G. response).

> *"Service-learning in rural education has been eye-opening for me. Before this experience, I underestimated the impact of socio-economic disparities and cultural differences on teaching and learning outcomes. Engaging directly with locals has shown me the importance of recognizing and honoring the assets present in rural communities while addressing the challenges they face. I now understand the significance of contextually relevant pedagogy and the role it plays in creating inclusive learning environments. This experience has deepened my commitment to incorporating inclusive teaching practices into my future STEM endeavors"* (student C. response).

Service-learning experiences prompted STEM students to critically reflect on their role as educators and agents of change [40,41]. They recognized the value of collaborative problem-solving, community-based partnerships, and asset-based approaches to education. Challenges encountered during service-learning, such as limited resources and systemic barriers, fueled participants' determination to advocate for equitable access to education and foster a sense of belonging for all students.

Service-learning immersed STEM students in real-world educational contexts, providing firsthand experiences of the challenges and opportunities present in rural communities. By engaging directly with students, educators, and community members, students gained insights into the diverse needs, backgrounds, and learning styles of rural learners [42]. This experiential learning approach fostered empathy, cultural humility, and a deep appreciation for the assets and complexities of rural education (Figure 2).

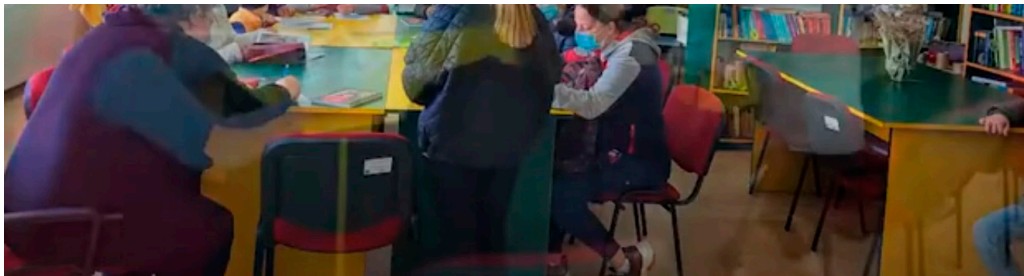

**Figure 2.** Storytelling club activities.

> *"Service learning has opened my eyes to the rich tapestry of rural education. By immersing myself in these communities, I've gained a newfound appreciation for the diverse backgrounds and learning styles of rural students. Engaging directly with educators and community members has helped me understand the importance of culturally responsive*

*teaching practices. This experience has taught me that effective STEM education isn't just about transferring knowledge; it's about building relationships and empowering students within their own context"* (student G. response).

*"Participating in service learning has been an invaluable experience for me as a STEM student. It's given me the opportunity to step outside of the classroom and into the real-world context of rural education. Through interactions with students and educators, I've gained insights into the unique challenges and opportunities present in these communities. This hands-on experience has fostered empathy and a deeper understanding of the cultural dynamics at play in education. It reinforced my belief in the importance of adapting teaching approaches to meet the needs of diverse learners"* (student A. response).

Cultural Awareness and Sensitivity: Service-learning exposed STEM students to the cultural, socio-economic, and linguistic diversity inherent in rural communities. Through meaningful interactions and dialogue, students developed a heightened awareness of the cultural norms, traditions, and values that shaped students' experiences and identities. They learned to recognize and respect cultural differences, incorporating culturally relevant pedagogy and materials into their teaching practices to create inclusive learning environments.

*"The project broadened my perspective. I realized the importance of acknowledging and addressing gender disparities in STEM fields. Hearing the stories of women who've overcome challenges in their careers made me more aware of the need for equal opportunities. It's not just about technical skills; it's about creating an environment where everyone feels empowered to contribute"* (student B. response).

Community Collaboration and Partnership: Service-learning encouraged STEM students to collaborate with local stakeholders, including teachers, parents, community leaders, and non-profit organizations. Through collaborative projects and initiatives, students co-created solutions to address educational disparities and promote equity in rural education. By engaging in dialogue and problem-solving with community members, students developed a deep understanding of community needs and aspirations, forging lasting partnerships that extended beyond the classroom.

*"The activities had a positive impact on community empowerment. We engaged with local schools, conducting STEM workshops for young girls. It was incredibly rewarding to see their enthusiasm and curiosity. By providing these opportunities, we're contributing to the empowerment of future generations of women in STEM"* (student G. response).

*"I believe that by fostering interest and skills in STEM from a young age, we're laying the groundwork for a more inclusive and diverse future in our community. These girls now have role models and mentors to look up to, and that can have a ripple effect on the entire community"* (student A. response).

*"I'm considering organizing a recurring STEM fair in collaboration with local schools, where we can showcase the exciting possibilities in STEM careers and provide mentorship opportunities. It's a small step, but I believe in the cumulative impact of such initiatives on breaking down barriers for women in STEM"* (student B. response).

Reflective Practice and Critical Inquiry: Service-learning encouraged STEM students to engage in reflective practice and critical inquiry, examining their assumptions, biases, and learning edges. Through reflective journals, group discussions, and feedback sessions, students critically evaluated their experiences, identified areas for growth, and refined their teaching approaches accordingly. This reflective process enabled students to challenge conventional paradigms, embrace diversity, and advocate for inclusive policies and practices within educational institutions.

*"The service-learning activities were eye-opening for me. I got the chance to develop not only my technical skills in STEM but also my interpersonal skills. Collaborating with women from diverse backgrounds and working towards a common goal allowed me to grow both personally and professionally"* (student L. response).

Adaptive Pedagogy and Differentiated Instruction: Service-learning prompted STEM students to develop adaptive pedagogical strategies that accommodated the diverse needs and abilities of rural learners. By employing differentiated instruction techniques, such as tiered assignments, flexible grouping, and scaffolding, students tailored their teaching methods to meet individual student needs and preferences. They leveraged technology, hands-on activities, and project-based learning to engage students actively in the learning process, fostering a culture of inclusion and accessibility (Figure 3).

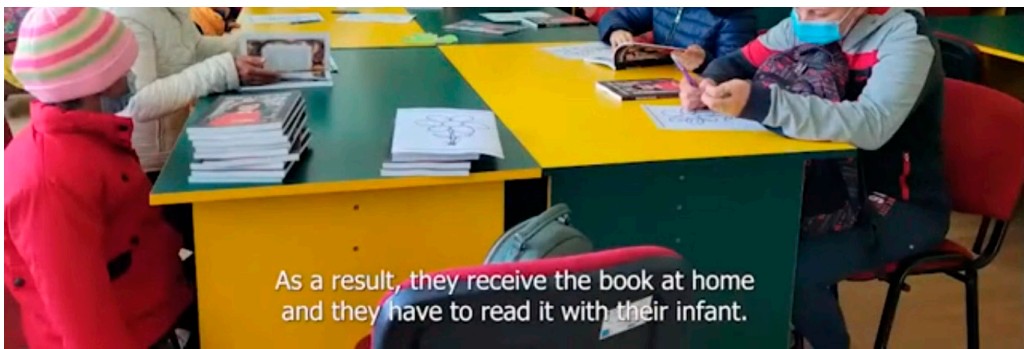

**Figure 3.** Adaptive pedagogy and differentiated instruction service-learning activities.

*"Storytelling was a powerful aspect of our project. It allowed us to humanize the statistics and put faces to the challenges women in STEM face. By sharing personal narratives, we could connect with our community on a more emotional level. It wasn't just about the numbers; it was about the individuals behind those numbers"* (student F. response).

Social Justice and Advocacy: Service-learning instilled in STEM students a sense of social justice and advocacy, inspiring them to become agents of change within their communities. By confronting systemic inequities and advocating for educational reform, students challenged the status quo and strove to create more equitable learning opportunities for all students. They recognized the importance of addressing root causes of educational disparities, including poverty, discrimination, and lack of access to resources, in fostering inclusive teaching practices and promoting educational equity.

*"There was a woman in the community who shared her journey in pursuing a STEM career despite societal expectations. Her resilience and passion for her work were inspiring. It made me realize the importance of advocating for equal opportunities and challenging stereotypes in our field"* (student B. response).

*"These activities have fueled my passion for advocating for diversity and equal opportunities in STEM. I see myself actively participating in mentorship programs and outreach initiatives to inspire more women to pursue careers in STEM. The experience has given me a sense of responsibility to contribute to positive change in the field"* (student G. response).

Experiences of service-learning shape STEM students' understanding of inclusive teaching practices by fostering cultural awareness, community collaboration, reflective practice, adaptive pedagogy, and social justice advocacy. Through meaningful engagement with rural communities, students develop the knowledge, skills, and dispositions needed to create inclusive learning environments that empower all students to succeed.

The research underscored the transformative potential of service-learning in preparing STEM students for future roles as inclusive educators in rural communities. By immersing themselves in authentic, experiential learning experiences, students developed the empathy,

cultural humility, and adaptive pedagogical strategies necessary for fostering inclusive learning environments. Moving forward, integrating service-learning into teacher preparation programs could enhance the capacity of STEM educators to address the diverse needs and aspirations of rural learners, ultimately advancing educational equity and social justice.

## 5. Discussion

The findings of this research contribute to the growing body of literature on innovative strategies for promoting gender equality in challenging socio-economic environments. The implications of this study extend beyond academia, informing policy, community development practices, and educational curricula tailored for rural settings. Ultimately, this research underscored the transformative potential of service-learning and storytelling as catalysts for positive change in the pursuit of gender equality in rural communities.

The integration of service-learning and storytelling strategies in STEM education has significant implications for STEM students. By intertwining these approaches, students not only engage in practical applications of their knowledge but also immerse themselves in narratives that bring relevance and context to their learning experiences. This combination fosters a deeper understanding of STEM concepts by connecting theoretical knowledge with real-world scenarios, enhancing students' ability to grasp complex ideas and apply them effectively. Additionally, the incorporation of service-learning encourages students to actively participate in their communities, promoting civic engagement, social responsibility, and a sense of purpose in their educational journey.

Service-learning allows STEM students to engage directly with rural communities, understand their unique challenges, and co-create solutions that promoted gender equality [43]. Through hands-on projects, students develop innovative technologies, educational programs, and community initiatives tailored to address the specific needs and aspirations of women and girls in rural areas.

Storytelling serves as a powerful tool for amplifying voices, raising awareness, and inspiring action. By sharing stories of resilience, empowerment, and achievement, STEM students challenge traditional narratives, highlight the contributions of women in rural communities, and spark meaningful conversations about gender equity and social change [44,45]. Additionally, storytelling can be used to make complex STEM concepts more accessible and meaningful by weaving narratives that connect theoretical knowledge to practical scenarios [24,27]. This approach helps students develop a deeper understanding of STEM subjects by providing context and relevance to their learning experiences.

Together, service-learning and storytelling create opportunities for dialogue, collaboration, and collective action towards a more just and equitable society. As STEM students engage in this process, they not only deepen their understanding of gender dynamics but also cultivate empathy, leadership skills, and a commitment to social justice that will shape their future endeavors and contribute to lasting positive impact in rural communities and beyond [46].

Service-learning and storytelling are powerful tools for improving culturally responsive teaching in rural education settings. By recognizing and valuing cultural backgrounds, embracing asset-based approaches, implementing differentiated instruction, fostering community-based learning, applying Universal Design for Learning principles, promoting teacher collaboration and professional development, and advocating for access to resources and support services, educators are able to create inclusive learning environments that honor the diverse strengths and identities of rural students.

Through storytelling, students listen to and share narratives that highlight the richness and significance of these cultural aspects. By actively recognizing and valuing cultural diversity, students develop a deeper appreciation for the unique identities and perspectives of rural students. Also, students amplify the voices and experiences of community members, showcasing their strengths, talents, and contributions. By highlighting these assets, educators are able to create more empowering and inclusive learning environments that celebrate the diverse strengths of rural students.

By listening to the stories of individual students and community members, educators gain insights into the unique needs and preferences of their students. This understanding informs the implementation of differentiated instruction strategies that cater to the diverse learning needs of rural students, ensuring that every student has equitable access to learning opportunities.

Through service-learning projects and storytelling initiatives, students engage in meaningful interactions with community members, gaining firsthand experiences and insights into local issues and challenges. This experiential learning fosters a deeper understanding of community dynamics and cultivates empathy and respect for rural perspectives.

Service-learning and storytelling align with the principles of Universal Design for Learning (UDL) by advocating for the design of learning experiences that are accessible and inclusive for all students.

Through collaborative initiatives, educators share best practices, resources, and strategies for promoting culturally responsive teaching in rural education settings. By engaging in reflective dialogue and storytelling sessions, educators continuously enhance their cultural competence and effectiveness in meeting the needs of rural students.

Service-learning initiatives and storytelling efforts also serve as platforms for connecting rural students and communities with essential resources and support services. Through service-learning experiences, students step out of the confines of the university campus and into rural educational settings. They encounter firsthand the challenges and opportunities present in rural schools, gaining practical insights into the realities of teaching in these communities.

## 6. Conclusions

This research underscored the value of service-learning as a transformative pedagogical approach for preparing STEM students for future teaching roles in rural communities. By engaging directly with local stakeholders and addressing authentic community needs, students developed a deep understanding of the complex interplay between education, culture, and social justice. Moving forward, continued investment in service-learning initiatives could empower STEM students to become catalysts for positive change, promoting inclusive practices and enhancing educational opportunities for all learners in rural contexts, as in other researchers' experiments [27]. This approach enriches the educational experience for STEM students, equipping them with not only technical skills but also a broader perspective on the societal impact and ethical dimensions of their field.

Innovative approaches to service-learning as a pedagogical strategy are vital for promoting equity and accessibility in rural education, especially for STEM students. These methods play a crucial role in fostering inclusive teaching practices that cater to the diverse needs of students in rural areas. By integrating service-learning into STEM education, educators can create opportunities for students to apply their scientific knowledge in real-world contexts, engage with their communities, and gain a deeper understanding of societal challenges. This hands-on approach not only enhances academic learning but also cultivates a sense of social responsibility and awareness among STEM students.

Furthermore, service-learning as a pedagogical approach in rural STEM education helps bridge the gap between theory and practice, making learning more relevant and impactful for students. By involving STEM students in community-based projects and initiatives, educators can address local STEM-related issues, promote sustainability, and empower students to actively contribute to solving real-world problems. This approach not only enriches the STEM learning experience but also nurtures essential skills such as critical thinking, problem-solving, and collaboration, which are vital for success in STEM fields.

Moreover, emphasizing service-learning in rural STEM education creates a more inclusive and supportive learning environment that meets the unique needs of STEM students. By tailoring teaching practices to address the challenges faced by rural schools, such as limited resources and diverse student populations, educators can promote STEM engagement, cultural understanding, and community involvement among students. Through

service-learning, STEM students can develop a deeper appreciation for the practical applications of their scientific knowledge, fostering a sense of purpose and connection to their communities.

So, as seen from the data collected, service-learning activities can address specific elements that are difficult to address in traditional university activities. By involving STEM students, they can become more sensitive to the following:

○ *Culturally Responsive Teaching*: Recognizing and valuing the cultural backgrounds, traditions, and experiences of rural students is fundamental to creating inclusive learning environments. Culturally responsive teaching involves incorporating students' cultural references, perspectives, and languages into the curriculum [47]. It encourages educators to establish meaningful connections with students and their communities, fostering a sense of belonging and respect for diverse identities.

○ *Asset-Based Approaches*: Instead of focusing solely on deficits and challenges, asset-based approaches emphasize the strengths, resources, and talents present within rural communities. By acknowledging and leveraging these assets, educators can co-create relevant and meaningful learning experiences that are built upon students' existing knowledge and skills [48]. This empowers students to see themselves as active participants in their own learning journey, fostering a sense of agency and confidence.

○ *Differentiated Instruction*: Recognizing that students in rural communities have diverse learning styles, interests, and abilities, differentiated instruction tailors teaching strategies and learning activities to meet individual needs. Educators can employ a variety of instructional methods, such as hands-on experiments, project-based learning, cooperative group work, and technology-enhanced activities, to accommodate diverse learners and ensure equitable access to educational opportunities [49,50].

○ *Community-Based Learning*: Integrating community-based learning experiences into the curriculum allows students to connect classroom learning with real-world contexts and issues relevant to their rural communities [51]. By engaging in service-learning projects, field trips, internships, and collaborative research initiatives, students develop a deeper understanding of local challenges and opportunities, while also contributing to community development and social change [52,53].

○ *Universal Design for Learning (UDL)*: UDL principles advocate for designing learning experiences that are flexible, accessible, and inclusive for all students, regardless of their backgrounds or abilities. This involves providing multiple means of representation, expression, and engagement to accommodate diverse learning preferences and needs [54,55]. Educators can utilize technology, multimedia resources, interactive materials, and varied assessment methods to scaffold learning and remove barriers to participation.

○ *Teacher Collaboration and Professional Development*: Building collaborative partnerships among educators, administrators, families, and community stakeholders is essential for promoting equity and excellence in rural education. Professional development opportunities that focus on culturally responsive teaching, inclusive pedagogy, and trauma-informed practices empower educators with the knowledge, skills, and resources needed to support diverse learners effectively [56,57].

○ *Access to Resources and Support Services*: Ensuring equitable access to high-quality instructional materials, technology infrastructure, libraries, laboratories, and support services is critical for leveling the playing field in rural education. Schools and districts need to prioritize investments in resources that address the unique needs and challenges of rural communities, including transportation, internet connectivity, health care, nutrition, and mental health services [58,59].

By embracing pedagogical approaches that prioritize equity, access, and inclusion, educators can create learning environments where all students in rural communities have the opportunity to thrive academically, socially, and emotionally. These approaches not

only empower students to reach their full potential but also contribute to the collective well-being and prosperity of rural communities.

## 7. Limitations and Delimitations

The interview protocol was designed to gather comprehensive insights into the students' experiences, reflections, and perceptions regarding the service-learning activities focused on emphasizing the role of women in society and promoting equal opportunities, particularly in STEM disciplines. While interviews can be a valuable method for gathering insights from STEM students after participating in service-learning activities focused on the role of women in society and equal opportunities, there were certain limitations to consider.

Participants might have provided responses that they perceived as socially desirable rather than expressing their genuine thoughts and experiences. This bias could have led to an overestimation of positive outcomes and may not have accurately reflected the challenges or negative aspects of the service-learning activities (social desirability bias).

Interviews typically captured a snapshot of participants' experiences at a specific point in time. Long-term impacts and changes in participants' perspectives over an extended period might not have been fully captured, limiting the understanding of sustained effects (limited long-term insights). Despite these limitations, interviews remain a valuable qualitative research method for exploring the experiences and perceptions of STEM students participating in service-learning activities. Combining interviews with other research methods (like journals and participatory video) and employing rigorous research design could have helped mitigate some of these limitations and provided a more comprehensive understanding of the impact of such initiatives.

Participants often felt inclined to present themselves in a favorable light, potentially leading to biases or the withholding of negative experiences, a phenomenon known as social desirability bias.

Conducting interviews was found to be time- and resource-intensive. This limitation emerged particularly when dealing with many participants, which hindered the feasibility of achieving a comprehensive understanding due to the substantial investment of time and resources required.

The subjectivity of stories was acknowledged as a limitation. Participants' narratives were inherently subjective and may not have always accurately reflected reality. There was a tendency for participants to emphasize certain aspects of their experiences while omitting others, introducing potential biases into the narrative.

Interpreting and analyzing stories posed challenges due to subjectivity. Researchers' perspectives influenced the interpretation of narratives, potentially leading to different interpretations of the same story. This subjectivity introduced variability in the analysis process, impacting the reliability and consistency of findings.

Producing participatory videos presented technical challenges. Participants and researchers may have lacked the necessary technical skills and resources, which could have compromised the quality or accessibility of the videos. These technical limitations constrained the effectiveness of participatory video as a method for capturing and sharing experiences.

**Funding:** This work was supported by a grant from the National Program for Research of the National Association of Technical Universities—GNAC ARUT 2023.

**Institutional Review Board Statement:** The study was conducted in accordance with the Declaration of Helsinki and approved by the Institutional Review Board (or Ethics Committee) of National University of Science and Technology POLITEHNICA of Bucharest. The ethical approval for the study was obtained from the Institutional Review Board, approval number 10322/05.10.2023.

**Informed Consent Statement:** Informed consent was obtained from all subjects involved in the study.

**Data Availability Statement:** Due to confidentiality agreements with the participants, this study's data are available only upon request from the author.

**Acknowledgments:** We are grateful to the students at University POLITEHNICA of Bucharest for the survey responses and carrying out activities with responsibility and respect for diversity.

**Conflicts of Interest:** The author declares no conflicts of interest. The funders had no role in the design of the study; in the collection, analyses, or interpretation of data; in the writing of the manuscript; or in the decision to publish the results.

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
