# Peer review of "Bridging Horizons: Exploring STEM Students’ Perspectives on Service-Learning and Storytelling Activities for Community Engagement and Gender Equality"

_2813-4346, doi:10.3390/higheredu3020020_

Round 1
Reviewer 1 Report
Comments and Suggestions for Authors
Specific remarks:
p.1.9 “Such an approach of equalizing genders can bring better conditions for their children…” suggestion “Such approach can bring better conditions for their children…”
p.1.24 An explanation of service learning should be provided.
p.2.48 is it necessary to make a new paragraph?
p.3.99-101 There is some repetition with what you said in lines 87, 88.
p.3.112_114 Is figure 1 adapted or is it from referenced authors? If it is from an author, you must ask permission to use the image. You should improve the image so that the information can be read.
p.3.116 “The STEM approach represents a paradigm shift in education…” but in line 84 we have the statement “The STEM approach …is an interdisciplinary method of education (line 84)”. The text should be clearer.
p.3. 118 – 155 What is the connection between storytelling approach and the STEAM approach? There is a lack of coherence between the previous text and the following one (storytelling approach)
p.5. 199 “STEM students participating in service-learning activities in rural communities…” What it means?
The research aims to investigate the impact of a combined service learning and storytelling project on the role of women in the school community. The study focuses on validating inclusive practices that emphasize the importance of women's contributions and equal opportunities.
P5. 202 Clarify how the triangulation was carried out.
P6. 249 - 379 “Results” It is not clear how the interviews were analysed. The evidence presented is from the same participant? Or not?...Confusing text. Should be improved. Evidence must be presented to support the claims made.
The discussion allows us to realize that the focus of the study is the transformative potential of service learning and storytelling as catalysts for positive change in the pursuit of gender equality in rural communities.
P6. 254 – xxx Where is the evidence to make the following claim? “Participants express a heightened awareness of the socio-economic disparities and 254 cultural nuances inherent in rural communities, underscoring the importance of contextually relevant pedagogy. Engaging directly with local stakeholders fosters empathy, cultural competency, and a deep appreciation for the assets and challenges present in rural educational settings”.
p.7 281- 286 The image must ensure the anonymity of those involved. A child's face is seen, Attention to ethical issues.
p.10 -11 “Discussion” and “Conclusions”, I suggest that you improve the text, basing it on the empirical study and not elaborate ideas, which although they are important, not all have sustainability in the results presented.
What does this mean for STEM students? Although they have presented the STEM method, it is not clear how it is connected to the empirical study they have developed.
p.References The hyphen needs to be replaced by an en dash (between the page numbers).
Submission Date
17 fev. 2024

Author Response
p.1.9 “Such an approach of equalizing genders can bring better conditions for their children…” suggestion “Such approach can bring better conditions for their children…”. The manuscript was adapted.
p.1.24 An explanation of service learning should be provided. In the manuscript it was included service learning details.
p.2.48 is it necessary to make a new paragraph? The manuscript was adapted.
p.3.99-101 There is some repetition with what you said in lines 87, 88. The manuscript was adapted.
p.3.112_114 Is figure 1 adapted or is it from referenced authors? If it is from an author, you must ask permission to use the image. You should improve the image so that the information can be read. Is made by the author and adapted.
p.3.116 “The STEM approach represents a paradigm shift in education…” but in line 84 we have the statement “The STEM approach …is an interdisciplinary method of education (line 84)”. The text should be clearer. The manuscript was adapted.
p.3. 118 – 155 What is the connection between storytelling approach and the STEAM approach? There is a lack of coherence between the previous text and the following one (storytelling approach). In the manuscript the paragraph needed was writen
p.5. 199 “STEM students participating in service-learning activities in rural communities…” What it means?
The research aims to investigate the impact of a combined service learning and storytelling project on the role of women in the school community. The study focuses on validating inclusive practices that emphasize the importance of women's contributions and equal opportunities.
In the manuscript was included: The Storytelling Club is a service-learning initiative aimed at challenging traditional societal roles and empowering women from rural communities to become active partici-pants in shaping their futures beyond traditional roles such as housekeeping. The club brings together women of diverse ages (from 19-46 years) and backgrounds to share their stories, experiences, and aspirations through storytelling sessions. The primary objective of the club is to provide a space for rural women to amplify their voices, share their expe-riences, and explore alternative pathways beyond conventional gender roles. By fostering a supportive and inclusive environment, the club aims to empower women to pursue ed-ucational and career opportunities, engage in community development initiatives, and actively participate in their children's education. The students engaged in a variety of ac-tivities that integrated STEM principles and concepts within the context of service learning as storytelling sessions and skill-building workshops. Regular storytelling sessions were held where women shared personal narratives, anecdotes, and life experiences. These ses-sions provided a space for self-expression, reflection, and mutual support among participants. The club organized workshops focused on developing practical skills such as pub-lic speaking, writing, and digital literacy. These workshops aimed to enhance partici-pants' confidence and empower them to articulate their thoughts and ideas effectively.
The aim of this research is to examine the influence of service-learning experiences on STEM students' perspectives and readiness for future teaching positions in rural commu-nities. Through qualitative analysis, the study delves into the reflections, insights, and transformative encounters of STEM students involved in service-learning projects de-signed to tackle educational obstacles in rural environments
P5. 202 Clarify how the triangulation was carried out. In the manuscript was included the paragraph: Triangulation was used as a methodological approach that involved the systematic inte-gration of data from multiple sources to enhance the credibility and validity of qualitative findings within a specific educational context like the service learning. By incorporating diverse research methods (semi-structured interviews, reflective journals, participatory video) and data collection techniques, triangulation served to provide a more comprehen-sive and reliable analysis of the subject under investigation. This enabled them to trian-gulate different perspectives, data sources, and methodologies to strengthen the trustwor-thiness of their interpretations and deepen their understanding of complex educational phenomena.
P6. 249 - 379 “Results” It is not clear how the interviews were analysed. The evidence presented is from the same participant? Or not?...Confusing text. Should be improved. Evidence must be presented to support the claims made.
The discussion allows us to realize that the focus of the study is the transformative potential of service learning and storytelling as catalysts for positive change in the pursuit of gender equality in rural communities.
The manuscript was adapted.
P6. 254 – xxx Where is the evidence to make the following claim? “Participants express a heightened awareness of the socio-economic disparities and 254 cultural nuances inherent in rural communities, underscoring the importance of contextually relevant pedagogy. Engaging directly with local stakeholders fosters empathy, cultural competency, and a deep appreciation for the assets and challenges present in rural educational settings”.
In the newest version was included.
p.7 281- 286 The image must ensure the anonymity of those involved. A child's face is seen, Attention to ethical issues. In the newest version was adapted.
p.10 -11 “Discussion” and “Conclusions”, I suggest that you improve the text, basing it on the empirical study and not elaborate ideas, which although they are important, not all have sustainability in the results presented.
What does this mean for STEM students? Although they have presented the STEM method, it is not clear how it is connected to the empirical study they have developed.
In the newest version was adapted.
p.References The hyphen needs to be replaced by an en dash (between the page numbers).
In the newest version was adapted.
Reviewer 2 Report
Comments and Suggestions for Authors
Bridging Horizons: Exploring STEM Students' Perspectives on Service-Learning and Storytelling Activities for Community Engagement and Gender equality
Line 7 & 11: sentence lacks a subject.
Line 40: Sentence needs revision or could be removed, and the rural context discussed later on.
Line 51-61: Good explanation regarding the importance of addressing gender inequity.
Lines 76-78: this sentence feels like repeated information and can be removed.
Line 265-269: This is the study rationale, which should come earlier in the paper.
Line 278-280: This quote lacks context and does not relate to “experiential learning.”
Questions:
· What is the location or region of the study?
· There is not a stated research question(s).
· How many participants were there?
· What was the interaction of participants with students in the study?
· What types of activities did the students engage in? STEM is discussed, but it is unclear how the activities were part of the service learning. What is the “innovative strategy” implemented? Is it the storytelling method? If so, this needs to be made more clearly how storytelling was utilized.
Comments:
· The research method is appropriate to examine impacts of service learning on STEM Students perceptions and preparedness for future teaching in rural communities, but the paper lacks citations and references for the methodological choices and provide justification for the approach to the research method.
· Some of the interview questions (page 6) are worded in such a way that might solicit “yes” or “no” responses, vs. more elaborative responses.
· The results should be stated in past tense.
· There is not an introduction to the results. For example, the themes are not included to orient the reader to what the findings are.
· Photos should have faces of children blurred.
· Several of the quotes are out of context.
· The results read similarly to a literature review because they are not in the correct tense.
· Many great approaches are mentioned in the conclusion, but it is unclear how this is reflected within the study itself.
· In the limitations, the small sample of participants is not an issue in qualitative method, since the intention is to build theory rather than generalizability. This should be revised to connect to the storytelling method and power of lived experiences, which is not a limitation, but a strength of the study.
Comments on the Quality of English LanguageThe tense needs to be corrected in the results section. In the current form, it is difficult to determine what occured in the study and what is part of the current literature.
Author Response
Line 7 & 11: sentence lacks a subject. In the newest version was adapted.
Line 40: Sentence needs revision or could be removed, and the rural context discussed later on. In the newest version was adapted.
Line 51-61: Good explanation regarding the importance of addressing gender inequity. Thanks!
Lines 76-78: this sentence feels like repeated information and can be removed. In the newest version was adapted.
Line 265-269: This is the study rationale, which should come earlier in the paper. In the newest version was adapted.
Line 278-280: This quote lacks context and does not relate to “experiential learning.” In the newest version was adapted.
Questions:
- What is the location or region of the study? In the newest version was included.
- There is not a stated research question(s).
- How many participants were there? In the newest version was included.
- What was the interaction of participants with students in the study? In the newest version was included.
- What types of activities did the students engage in? STEM is discussed, but it is unclear how the activities were part of the service learning. What is the “innovative strategy” implemented? Is it the storytelling method? If so, this needs to be made more clearly how storytelling was utilized.
In the newest version was included.
Comments:
- The research method is appropriate to examine impacts of service learning on STEM Students perceptions and preparedness for future teaching in rural communities, but the paper lacks citations and references for the methodological choices and provide justification for the approach to the research method. In the newest version was adapted.
- Some of the interview questions (page 6) are worded in such a way that might solicit “yes” or “no” responses, vs. more elaborative responses.
- The results should be stated in past tense. In the newest version was adapted.
- There is not an introduction to the results. For example, the themes are not included to orient the reader to what the findings are. In the newest version was included.
- Photos should have faces of children blurred. In the newest version was included.
- Several of the quotes are out of context. In the newest version was included.
- The results read similarly to a literature review because they are not in the correct tense. In the newest version was adapted.
- Many great approaches are mentioned in the conclusion, but it is unclear how this is reflected within the study itself. In the newest version was adapted.
- In the limitations, the small sample of participants is not an issue in qualitative method, since the intention is to build theory rather than generalizability. This should be revised to connect to the storytelling method and power of lived experiences, which is not a limitation, but a strength of the study. Many thanks!
Round 2
Reviewer 1 Report
Comments and Suggestions for Authors
This new version of the work includes the suggestions made in the first revision.
-removed for peer-review-
Author Response
Thanks for accept!
Reviewer 2 Report
Comments and Suggestions for Authors
· Abstract is improved!
· In line 45: there could be a definition of service learning here. It is important to include earlier than in line 195.
· Line 137-144: Great paragraph about the connection between storytelling and STEM but needs a citation within the paragraph. I see there are some articles cited in the following paragraph, but it would help to lead with these research findings in other contexts.
· Lines 146-157: this is also good information, but it might be shortened to be more concise. Lines 153-157 feels a little redundant.
· I still think this objective needs to be asked in the form of a question. “What is the impact of…?” and “What are the STEM student perceptions of…?”
· Line 280: make sure this is in past tense.
· Line 284: rationale for who was included needs to be earlier. Then the section can indicate the number of participants and setting. Please indicate if the 12 women also completed consent procedures.
· Line 290 Start the sentence with word form of numeral. And women is plural form.
· The analysis plan is missing. How was the data analyzed?
· On pages 10-14, the quotes are important, but need more context and explanation. Without an understanding of the analysis, this is limited.
· The change of the tense in the discussion is appropriate.
· The elements that are discussed on pages 18 and 19 should be integrated into the discussion.
· Line 885: the purpose of interviews is in depth data and insight not generalization. There needs to be a better presentation of the method, as the limitations are currently described quantitively.
· Lines 897-905: this makes more sense in terms of the limitation of the study.
Comments on the Quality of English LanguageThe English is improved. Only small grammatical changes suggested.
Author Response
- In line 45: there could be a definition of service learning here. It is important to include earlier than in line 195. The manuscript was updated.
Service-learning is a transformative pedagogy that engages students, community members, and instructors in co-creating partnerships to address community challenges [1] [2]. It is grounded in active and experiential learning principles, advocating for comprehensive education that includes academic, civic, and personal growth [3]. Service-learning empowers students to identify and solve community problems, fostering personal and social transformation [4]. By integrating service-learning into higher education, institutions can enhance academic learning, civic engagement, and personal development while promoting democratic partnerships with communities [5]. The pedagogical strategy of service-learning not only enriches educational experiences but also contributes to the advancement of student outcomes across various domains, making it a valuable tool for educational innovation and social change.
- Line 137-144: Great paragraph about the connection between storytelling and STEM but needs a citation within the paragraph. I see there are some articles cited in the following paragraph, but it would help to lead with these research findings in other contexts. The manuscript was updated.
The connection between the storytelling approach and the STEM (Science, Technology, Engineering, and Mathematics) approach lies in how storytelling can enhance engagement, understanding, and retention of STEM concepts among students. Storytelling in STEM education serves as a powerful pedagogical tool that makes complex scientific information more approachable, relatable, and meaningful to learners, from early ages [25,26]. By incorporating narratives, anecdotes, and real-world examples into STEM les-sons, educators can create educational environments where students develop emotional connections to the subject matter, fostering deeper engagement and interest in STEM disciplines.
Research and articles [24,25,26] highlight the benefits of using storytelling in STEM education to communicate science effectively, engage students in problem-solving activities, and enhance their critical thinking skills. Storytelling not only makes STEM content more accessible but also helps students connect theoretical knowledge to practical applications, promoting a deeper understanding of scientific concepts.
The integration of storytelling in STEM education aligns with the goal of making STEM learning more engaging, relevant, and impactful for students. By weaving narratives into STEM instruction [27], educators can create a dynamic learning environment that fosters creativity, critical thinking, and a deeper appreciation for the interconnected-ness of science, technology, engineering, and mathematics in real-world contexts.
- Lines 146-157: this is also good information, but it might be shortened to be more concise. Lines 153-157 feels a little redundant. The manuscript was updated.
- I still think this objective needs to be asked in the form of a question. “What is the impact of…?” and “What are the STEM student perceptions of…?”
Research questions:
- What is the impact of service-learning experiences on STEM students' attitudes, beliefs, and preparedness for future teaching roles in rural communities?
- How do service-learning initiatives in rural settings shape the perspectives on teaching and community engagement of STEM students, as reflected in their insights and reflections?
- Line 280: make sure this is in past tense. .The manuscript it was updated.
- Line 284: rationale for who was included needs to be earlier. Then the section can indicate the number of participants and setting. Please indicate if the 12 women also completed consent procedures. The manuscript was updated.
The women participants in the storytelling club belonged to a broad spectrum of ages, ranging from 19 to 46 years old. The twelve numbers of this category of participants contributed to a rich exchange of perspectives (diversity in age), with members representing different life stages, viewpoints, and experiences, as the community diversity.
In the research study, twenty-one STEM student participants engaged in storytelling activities as part of the research methodology. These students were actively involved in various STEM disciplines, spanning science, technology, engineering, and mathematics.
The research was conducted in a rural area, Calarasi county, Curcani, Romania. Twelve women from this area and their children were the target group of the ser-vice-learning activities carried out by students from the Polytechnic University of Bucharest. The community of Curcani was selected to participate in the service-learning activity based on several factors, with the primary consideration being the expressed interest and advocacy from one of the STEM students who hailed from this community. This student, who was intimately familiar with the challenges and needs of Curcani, voiced a strong desire to contribute to the betterment of their hometown and to raise awareness of its necessities.
- Line 290 Start the sentence with word form of numeral. And women is plural form. . In the manuscript it was updated.
- The analysis plan is missing. How was the data analyzed? The manuscript was updated.
To assess the impact of service-learning experiences on STEM students' attitudes and preparedness for teaching roles in rural communities, a comprehensive analysis plan was devised. This plan included conducting semi-structured interviews with students to explore their reflections on service-learning activities, learning and skill development, im-pact on perceptions and attitudes, community interaction and engagement, recommendations and future involvement, and personal connection and empathy. Reflective journaling was utilized to capture personal and professional growth insights. Additionally, participatory video creation was facilitated to showcase their experiences and perspectives.
Furthermore, the reflections and insights of STEM students engaged in ser-vice-learning initiatives in rural settings were investigated to provide valuable information on how these experiences shaped their views on teaching and community engagement.
For qualitative analysis, thematic analysis was used to identify key themes emerging from interviews, journal entries, and participatory videos. This involved looking for pat-terns in attitudes, beliefs, and preparedness for teaching roles, as well as comparing and contrasting reflections to understand the impact of service-learning on students' perspectives.
For cross-data validation, findings from interviews, journal entries, and videos were tri-angulated to validate themes and insights. Consistency and discrepancies across all data sources were examined to enhance the credibility of the analysis.
Triangulation was used as a methodological approach that involved the systematic integration of data from multiple sources to enhance the credibility and validity of qualitative findings within a specific educational context like service learning. By incorporating diverse research methods (semi-structured interviews, reflective journals, participatory video) and data collection techniques, triangulation served to provide a more comprehensive and reliable analysis of the subject under investigation. This enabled them to triangulate different perspectives, data sources, and methodologies to strengthen the trustworthiness of their interpretations and deepen their understanding of complex educational phenomena.
In the research context, the study was conducted at the largest STEM university in the country, situated in the heart of a bustling cosmopolitan city. This university attracted students from a wide array of backgrounds and cultures. These selected students were well-accustomed to the fast-paced urban lifestyle prevalent in the city, characterized by towering skyscrapers, bustling nightlife, and a rich tapestry of languages and traditions. Most of these students had spent their formative years in urban environments, navigating the intricacies of city life without significant exposure to rural communities. Through ser-vice-learning experiences, students stepped out of the confines of the university campus and into rural educational settings. They encountered firsthand the challenges and opportunities present in rural schools, gaining practical insights into the realities of teaching in these communities.
In rural areas, students encountered a rich tapestry of cultures, socio-economic backgrounds, and linguistic diversity that differed from their urban upbringing. This exposure broadened their perspectives and fostered a deeper understanding of the unique needs and assets of communities.
Engaging in service-learning projects required collaboration and teamwork among STEM students, educators, and community members. This collaborative approach fostered a sense of camaraderie and collective responsibility towards addressing educational disparities in rural areas.
Service-learning prompted students to reflect critically on their own assumptions, biases, and preconceptions about rural education and community engagement. Through introspection and dialogue, students challenged their existing beliefs and cultivated a more nuanced understanding of inclusive teaching practices.
As students navigated the complexities of rural settings, they were compelled to adapt their pedagogical approaches to meet the diverse needs of their students. This involved implementing innovative teaching methods, leveraging technology, and incorporating culturally relevant content into their lesson plans.
Through service-learning experiences, students developed a heightened awareness of social injustices and inequities in education. They were motivated to advocate for systemic change and to leverage their STEM expertise to create more inclusive and equitable learning environments in rural communities.
- On pages 10-14, the quotes are important, but need more context and explanation. Without an understanding of the analysis, this is limited. The manuscript was updated.
- The change of the tense in the discussion is appropriate.
- The elements that are discussed on pages 18 and 19 should be integrated into the discussion. The manuscript was updated.
Service learning and storytelling were powerful tools for improving culturally responsive teaching in rural education settings. By recognizing and valuing cultural backgrounds, embracing asset-based approaches, implementing differentiated instruction, fostering community-based learning, applying Universal Design for Learning principles, promoting teacher collaboration and professional development, and advocating for access to resources and support services, educators were able to create inclusive learning environments that honored the diverse strengths and identities of rural students.
Through storytelling, students listened to and shared narratives that highlighted the richness and significance of these cultural aspects. By actively recognizing and valuing cultural diversity, students developed a deeper appreciation for the unique identities and perspectives of rural students. Also, students amplified the voices and experiences of community members, showcasing their strengths, talents, and contributions. By highlighting these assets, educators were able to create more empowering and inclusive learning environments that celebrated the diverse strengths of rural students.
By listening to the stories of individual students and community members, educators gained insights into the unique needs and preferences of their students. This understanding informed the implementation of differentiated instruction strategies that catered to the diverse learning needs of rural students, ensuring that every student had equitable access to learning opportunities.
Through service-learning projects and storytelling initiatives, students engaged in meaningful interactions with community members, gaining firsthand experiences and insights into local issues and challenges. This experiential learning fostered a deeper understanding of community dynamics and cultivated empathy and respect for rural perspectives.
Service learning and storytelling aligned with the principles of Universal Design for Learning (UDL) by advocating for the design of learning experiences that were accessible and inclusive for all students.
Through collaborative initiatives, educators shared best practices, resources, and strategies for promoting culturally responsive teaching in rural education settings. By engaging in reflective dialogue and storytelling sessions, educators continuously enhanced their cultural competence and effectiveness in meeting the needs of rural students.
Service-learning initiatives and storytelling efforts also served as platforms for connecting rural students and communities with essential resources and support services. Through service-learning experiences, students step out of the confines of the university campus and into rural educational settings. They encounter firsthand the challenges and opportunities present in rural schools, gaining practical insights into the realities of teaching in these communities.
- Line 885: the purpose of interviews is in depth data and insight not generalization. There needs to be a better presentation of the method, as the limitations are currently described quantitively. The manuscript was updated.
Participants often felt inclined to present themselves in a favorable light, potentially leading to biases or withholding of negative experiences, a phenomenon known as social desirability bias.
Conducting interviews was found to be time and resource intensive. This limitation emerged particularly when dealing with many participants, which hindered the feasibility of achieving a comprehensive understanding due to the substantial investment of time and resources.
The subjectivity of stories was acknowledged as a limitation. Participants' narratives were inherently subjective and may not have always accurately reflected reality. There was a tendency for participants to emphasize certain aspects of their experiences while omitting others, introducing potential biases into the narrative.
Interpreting and analyzing stories posed challenges due to subjectivity. Researchers' perspectives influenced the interpretation of narratives, potentially leading to different interpretations of the same story. This subjectivity introduced variability in the analysis process, impacting the reliability and consistency of findings.
Producing participatory videos presented technical challenges. Participants and researchers may have lacked the necessary technical skills and resources, which could have compromised the quality or accessibility of the videos. These technical limitations constrained the effectiveness of participatory video as a method for capturing and sharing experiences.
- Lines 897-905: this makes more sense in terms of the limitation of the study. The manuscript was updated.
Round 3
Reviewer 2 Report
Comments and Suggestions for Authors
Line 13-17
The first two sentences of the abstract might be condensed or combined. The third sentence of the abstract is very strong to convey the study.
Service-learning definition has been added earlier in the manuscript. This helps the reader understand the definition.
Citations added regarding the connection between storytelling and STEM.
The claim made on line 151-153 is referencing citation [24, 25, and 26}?
The research questions have been improved.
In the participant section, line. 292, are 12 women part of the total 21? The sentence on 303 “The twelve numbers of this category of “ might be revised to be clearer. Thank you for including consent procedures.
The data analysis is described in more detail.
Please consider making the discussion more succinct. There are several places where information feels repeated.
The limitations are improved. One way to navigate subjectivity is to address this in the analysis plan by having members of the research team to discuss the analysis and patterns to improve interpretation.
Author Response
Line 13-17
The first two sentences of the abstract might be condensed or combined. The third sentence of the abstract is very strong to convey the study.
It was improved:
This study explores STEM students' perspectives on service-learning and story-telling activities to enhance community engagement and advance gender equality, investigating their impact on students' perceptions, experiences, and understanding of gender dynamics within rural communities.
Service-learning definition has been added earlier in the manuscript. This helps the reader understand the definition. Many thanks for the suggestion.
Citations added regarding the connection between storytelling and STEM. Many thanks for the suggestion.
The claim made on line 151-153 is referencing citation [24, 25, and 26}? It was added.
The research questions have been improved. Many thanks for the suggestion
In the participant section, line. 292, are 12 women part of the total 21? The sentence on 303 “The twelve numbers of this category of “ might be revised to be clearer. Thank you for including consent procedures.
It was improved. Many thanks for the suggestion.
The data analysis is described in more detail.
Please consider making the discussion more succinct. There are several places where information feels repeated. It was improved. Many thanks for the suggestion
The limitations are improved. One way to navigate subjectivity is to address this in the analysis plan by having members of the research team to discuss the analysis and patterns to improve interpretation.
Many thanks for the suggestion. It was improved.
